# Chronic illness and extreme performance: Type 1 diabetes in ultra-endurance

Jean-Charles Vauthier[1,2,3]*, Lucie Choley[2], Bernard Kabuth[1,3]

1 Université de Lorraine, France, 2 Département de médecine Générale, Faculté de médecine, Degeresp, Nancy, France, 3 InterPSY UR, Université de Lorraine, Nancy, France

* jean-charles.vauthier@univ-lorraine.fr

## Abstract

### Background

This study explores how people with type 1 diabetes experience ultra-endurance sports, focusing on identity, self-management, and psychosocial impacts through a qualitative approach.

### Methods and findings

This qualitative study used a constructivist grounded theory approach. Thirteen semi-structured interviews were conducted with French-speaking adults living with T1D who had completed at least one marathon or ultra-endurance race. Data were analyzed inductively through line-by-line coding, focused coding, and thematic categorization, supported by NVivo® software and reported in accordance with COREQ guidelines.

Four major themes emerged: (1) a progressive process of acceptance, from the shock of diagnosis to identity integration; (2) the ambivalence of being "ill," with tensions between medical status and personal identity; (3) the mental load of diabetes management, intensified by ultra-endurance but mitigated by adaptive strategies and peer support; and (4) sport as a transformative space for self-affirmation, resilience, and advocacy. Participants described how ultra-endurance helped them reclaim agency, reframe their illness, and inspire others.

### Conclusions

Ultra-endurance sports offer a unique context for identity reconstruction and empowerment among people with T1D. Far from being a barrier, the illness can become a driver of personal growth and social visibility. These findings highlight the importance of recognizing experiential knowledge in chronic disease management and call for more inclusive, patient-centered approaches in healthcare and research.

**Data availability statement:** The anonymized transcripts underlying the findings of this study are available from Zenodo at DOI: https://doi.org/10.5281/zenodo.17741890.

**Funding:** The author(s) received no specific funding for this work.

**Competing interests:** NO authors have competing interests.

## Introduction

Type 1 diabetes (T1D) is a chronic autoimmune condition that requires continuous self-management, including insulin therapy, blood glucose monitoring, dietary regulation, and physical activity. This constant vigilance imposes a significant mental and emotional burden, particularly among young adults, and can affect psychological well-being, quality of life, and identity development [1–3].

While moderate physical activity is widely recommended in T1D management, the lived experience of individuals engaging in extreme physical challenges—such as ultra-endurance sports—remains underexplored. Ultra-endurance events, lasting hours or even days, create unique physical and psychological challenges. For individuals with T1D, these challenges include anticipating energy needs, adjusting insulin regimens in real time, and managing the risks of hypo- or hyperglycemia [4].

Existing literature on T1D and exercise has primarily focused on physiological outcomes, therapeutic adjustments, and metabolic benefits [5,6]. However, qualitative studies that explore the subjective experience of living with T1D in high-performance contexts are scarce. The few available studies suggest that sport can serve as a powerful medium for identity reconstruction, allowing individuals to shift from a "sick" identity to that of an "athlete" or "role model" [7,8].

Moreover, recent reviews have highlighted a lack of patient involvement in research on T1D and exercise, and a limited focus on experiential dimensions such as autonomy, emotional well-being, and self-perception [6]. These gaps are most evident where illness meets expectations of performance, self-transcendence, and visibility.

This study aims to explore the lived experience of individuals with T1D who engage in ultra-endurance sports. Using a qualitative approach grounded in constructivist grounded theory, we examine how these individuals negotiate the constraints of their condition, integrate the illness into their identity, and use sport as a space for resilience, transformation, and recognition. By centering the voices of patients, this research contributes to a more nuanced understanding of chronic illness in extreme contexts and invites a rethinking of the boundaries between health, performance, and vulnerability.

This study is part of a broader qualitative research project exploring the lived experience of individuals with type 1 diabetes in the context of endurance sports. While the present article focuses on individual trajectories and identity negotiation in ultra-endurance, other components of the project address complementary dimensions such as peer communities, patient–physician relationships, and the role of self-monitoring technologies. By centering the voices of patients, this study contributes to a more nuanced understanding of chronic illness in extreme contexts and invites a rethinking of the boundaries between health, performance, and vulnerability.

## Materials and methods

### Study design

This study employed a qualitative design based on constructivist grounded theory, as developed by Charmaz [9] This approach is particularly suited to exploring complex, subjective experiences and allows for the co-construction of meaning between researchers and participants.

To clarify the epistemological positioning and ensure methodological coherence, we also drew on Rieger's typology of grounded theory approaches, which distinguishes between objectivist, constructivist, and postmodern variants [10]. Our study aligns with the constructivist tradition, emphasizing reflexivity, contextual interpretation, and the co-production of knowledge.

The study followed the COREQ (Consolidated Criteria for Reporting Qualitative Research) guidelines to ensure transparency and rigor (see Supporting Information). The aim was to generate theoretical insights into how individuals living with type 1 diabetes (T1D) experience and manage their condition in the context of ultra-endurance sports.

### Researcher positioning

The principal investigator (JCV) occupies a dual position as both a healthcare professional and a person living with type 1 diabetes. This insider status facilitated access to participants and enabled nuanced understanding of their experiences. However, it also raised potential risks of interpretive bias and over-identification. To mitigate these risks, several strategies were implemented: maintaining a reflexive journal throughout data collection and analysis; systematically documenting emotional responses and methodological decisions; and engaging in regular debriefing sessions with co-authors who did not share the same experiential background. These measures aimed to ensure critical distance and enhance the credibility of the findings.

A reflexive journal was maintained throughout the research process to document methodological decisions, emotional responses, and evolving interpretations. This practice aligns with current recommendations for enhancing transparency and reflexivity in qualitative research [11].

Regular discussions with the research team (LC and BK), who did not share the same experiential background, contributed to analytical triangulation and helped ensure critical distance. This reflexive stance aligns with the constructivist grounded theory approach adopted in this study, which acknowledges the co-construction of meaning between researcher and participants.

### Sampling and recruitment

Participants were recruited using purposive sampling to ensure the inclusion of individuals with relevant experience. Eligibility criteria were: (I) age 18 or older; (II) living with type 1 diabetes; (III) having completed at least one marathon or ultra-endurance race (road or trail) in the past five years; and (IV) fluency in French and ability to provide informed consent.

Recruitment was conducted between July 4, 2024 and December 19, 2024, through diabetes peer-support groups, social media platforms, and word-of-mouth. The final sample consisted of 13 participants (11 men and 2 women), aged 26–66 years, with a mean diabetes duration of 18.7 years.

Sampling was adjusted iteratively during data collection and analysis, following the principles of theoretical sampling. Recruitment ceased when interpretive saturation was reached—that is, when no new conceptual insights emerged from additional interviews.

All participants provided written informed consent, signed in two copies: one retained by the participant and one by the researcher. Participants also consented to the sharing of anonymized transcripts for research purposes.. The study was approved by the South-East VI Research Ethics Committee (Reference: 24.00907.000253).

### Data collection

Data were collected through semi-structured interviews conducted via videoconference between July and December 2024. All interviews were conducted by the principal investigator (JCV), who shared experiential proximity with participants. Interviews were audio-recorded with consent and fully transcribed verbatim. Transcripts were enriched with contextual notes and non-verbal cues when relevant.

The interview guide was developed based on a review of the literature and refined iteratively during the study. It included open-ended questions exploring diagnosis, daily management, emotional and social experiences, and the role of sport in identity and self-perception.

Interviews lasted between 47 and 68 minutes (mean: 57 minutes). Audio files were deleted after transcription, in accordance with ethical requirements. All data were anonymized and securely stored.

Field notes and reflexive memos were written after each interview to capture impressions, emerging themes, and methodological reflections.

## Data analysis

Data were analyzed using constructivist grounded theory, following the procedures outlined by Charmaz [9]. The analysis involved several iterative steps: (I) line-by-line initial coding of transcripts; (II) focused coding to identify recurring patterns; (III) development of conceptual categories; and (IV) integration of themes through constant comparison.

Analytical memos were written throughout the process to document emerging insights, theoretical reflections, and methodological decisions. Coding and data organization were supported by NVivo® software (version 1.7.2).

The research team engaged in regular discussions to refine categories and ensure analytical coherence. Triangulation was achieved through collaborative interpretation among the three authors, who brought complementary perspectives to the analysis.

Sampling and analysis were conducted in parallel, and recruitment was stopped when interpretive saturation was reached—that is, when no new conceptual insights emerged from additional data [12,13].

The analysis emphasized reflexivity, contextual interpretation, and the co-construction of meaning, in line with the constructivist epistemology of the study.

## Ethical considerations

This study was approved by the South-East VI Research Ethics Committee (Reference: 24.00907.000253) on April 5, 2024. All participants received detailed information about the study's objectives, procedures, and confidentiality measures, and provided written informed consent prior to participation.

Participants were informed of their right to withdraw at any time without consequence. Data were anonymized during transcription, and all identifying information was removed. Audio recordings were deleted after transcription, and all data were securely stored in accordance with institutional and ethical guidelines.

The study adhered to the ethical principles outlined in the Declaration of Helsinki and followed best practices for qualitative research involving human participants.

## Results

### Participant characteristics and study overview

Thirteen interviews were conducted between July and December 2024 with individuals living with type 1 diabetes and engaged in endurance or ultra-endurance sports. Participants (coded S1 to S13) varied in age, diabetes duration, and athletic background. Most were men (n = 11), with a mean age of 42.8 years (SD = 10.8) and a mean diabetes duration of 18.7 years (SD = 9.8). All had completed at least one marathon or ultra-endurance race in the past five years. Interviews lasted between 47 and 68 minutes (mean: 57 minutes). Detailed participant characteristics are presented in Table 1.

### Analytical approach

Thematic analysis followed the principles of constructivist grounded theory. After line-by-line initial coding, focused coding was used to group significant segments into emerging categories. Analytical memos were written throughout the process to support theoretical development. NVivo® software (v1.7.2) was used to facilitate coding and data organization.

**Table 1. Characteristics of study participants.**

| ID | Sex | Age | Age at diagnosis | Duration of T1D | Treatment | CGM | HbA1c | Sports practiced | Max distance |
|---|---|---|---|---|---|---|---|---|---|
| S1 | M | 36 | 21 | 15 | Pump (Omnipod) | Dexcom | 7.8 | Ultra-trail, ski, tennis-marathon | 200 km |
| S2 | M | 26 | 15 | 11 | MDI (pens) | Libre 2 | N/A | Trail, triathlon, rugby-marathon | 80 km |
| S3 | M | 53 | 38 | 15 | Pump (Omnipod Dash) | Dexcom | N/A | Ultra-trail-marathon | 108 km |
| S4 | M | 48 | 25 | 23 | Pump (Omnipod, Diabeloop planned) | Dexcom G6 | 7.4 | Trail, marathon | 59 km |
| S5 | F | 49 | 23 | 27 | Pump | Dexcom | 5.8 | Trail, ultra cycling, Marathon, triathlon | 42 km |
| S6 | M | 41 | 19 | 22 | Pump | Dexcom | N/A | Trail-marathon | 50 km |
| S7 | M | 66 | 40 | 26 | MDI (pens) | Libre | 6.7 | Trail-marathon | 50 km |
| S8 | M | 37 | 13 | 24 | Pump | Dexcom | N/A | Trail-marathon | 60 km |
| S9 | M | 46 | 42 | 4 | MDI (pens) | Libre | N/A | Trail-marathon | 42 km |
| S10 | M | 38 | 10 | 28 | Pump | Dexcom | 6.4-6.5 | Trail-marathon | 100 km |
| S11 | M | 44 | 19 | 25 | Pump | Dexcom | 6-6.2 | Trail-marathon | 80 km |
| S12 | M | 26 | 23 | 3 | MDI (pens) | Libre | N/A | Trail-marathon | 42 km |
| S13 | F | 46 | 16 | 30 | Pump | Dexcom | N/A | Trail-marathon | 50 km |

This table summarizes demographic and clinical characteristics of participants, including age, diabetes duration, treatment type, technology use, and athletic background. HbA1c values were not systematically collected; they are reported only when spontaneously mentioned by participants during interviews, in accordance with the protocol approved by the ethics committee.

The research team engaged in regular discussions to refine categories and ensure analytical coherence. Triangulation was achieved through collaborative interpretation among the three authors. Sampling and analysis were conducted in parallel, and recruitment was stopped when interpretive saturation was reached.

From Shock to Meaning: The Journey Toward Acceptance

The diagnosis of type 1 diabetes is often experienced as a brutal biographical rupture, an existential shock. Several runners described initial reactions of disbelief, fear, or denial:

"I was devastated […] I thought: 'This is my last pancake.'" (S5)

"When I found out, I reacted very badly. I didn't want to change anything in my life." (S5)

For others, the diagnosis coincided with other destabilizing events, such as the COVID-19 lockdown:

"I got the news on Friday, right before the first lockdown. On Monday, I was hospitalized, then sent home alone with my pens and test strips." (S9)

Some testimonies reflect a sudden awareness of the disease's severity, sometimes through critical episodes:

"I lost consciousness, convulsed, my friends held me down… I woke up in a helicopter." (S3)

The shock also stems from dietary constraints or negative social representations:

"I enjoy eating, enjoying life… And then they talk to me about dieting. I said: 'No way! I'll adapt my insulin to my life, not the other way around.'" (S4)

Others describe a gentler entry into the disease, facilitated by family support:

"I was lucky to have a mother who's a nurse. She let me manage it from the start." (S8)

## Identity Disruption and the Ambivalence of Being "Ill"

This theme reveals a strong identity tension. While the medical reality of the disease is undeniable, its integration into personal identity is far from straightforward.

Some acknowledge their status as ill, linked to insulin dependence:

"I don't consider myself sick, but I am. We inject a product every day, and if we don't, we die." (S9)

Others reject the label or seek to distance themselves from the social image of illness:

"I'm not sick. I'm diabetic, that's all!" (S7)

"I've never considered myself sick. I don't like the word 'sick,' just like I don't like 'patient.'" (S13)

The distinction between "having a disease" and "being sick" is recurrent:

"I don't feel sick. I have a system." (S11)

"When I'm exercising, I don't feel sick. But for insurance companies, I am." (S5)

This ambivalence is sometimes expressed with humor or irony, as a way to reclaim self-definition:

"Diabetic but not sick!" (S13)

## Mental Load and Adaptation Strategies

The mental load associated with T1D is omnipresent in the runners' narratives. It manifests as constant vigilance, anticipation, and psychological fatigue.

### A Constant Presence

Diabetes requires continuous attention, from waking to bedtime. One participant compares it to an extra child:

"It's like a third child, always there, always, always." (S9)

Monitoring, insulin adjustments, and dietary management become daily reflexes:

"You think about it all the time, you check your numbers, your scores." (S1)

### Sport-Specific Anticipation and Logistics

Ultra-endurance adds another layer of complexity. Each outing requires meticulous preparation:

"Even for a short run, I check beforehand, adjust the pump rate, take a fruit paste." (S8)

### Psychological and Social Consequences

This constant vigilance leads to exhaustion and affects spontaneity and social life:

"The mental load is the biggest obstacle." (S2)

"Spontaneously grabbing your shoes and going out? That's no longer possible." (S12)

## Technologies: Relief or Overload?

Technological aids offer control but can also increase mental load:

"Before, I checked 6 to 8 times a day. Now I check constantly." (S10)

Some even revert to simpler systems:

"I stopped using the closed loop. It was an extra mental burden." (S13)

## Sport as a Lever for Transformation and Assertion

For many runners, sport is not just a health activity—it's a space for self-affirmation, resilience, and identity reconstruction.

"I don't do it to prove anything to others, but to prove to myself that it's possible." (S2)

## Visibility and Advocacy

Running becomes a space of visibility and sometimes activism:

"It's important to show that the disease doesn't stop us." (S5)

## Inspiration and Social Role

Some runners take on a role model status, especially for young diabetics:

"Parents told me it reassured them to see that it's possible." (S6)

## Self-Management Skills and Learning

Ultra-endurance requires deep knowledge of one's body and diabetes. Strategies are highly personalized and often learned through trial and error:

"It took me four years to find balance." (S1)

## Reversing the Meaning of Illness

For several runners, the disease paradoxically becomes a source of strength:

"My diabetes helped me strengthen my mental resilience." (S1)

## Discussion

### Contextualizing the results

This study highlights the complexity of living with type 1 diabetes (T1D) in the context of ultra-endurance sports. The diagnosis was frequently described as a biographical rupture, echoing Bury's concept of *biographical disruption* [14], where illness interrupts the continuity of identity and life trajectory. Participants' narratives also resonate with Claire Marin's reflections on illness as an existential fracture that forces a redefinition of self and relationships [15].

Beyond the initial shock, the process of acceptance emerged as dynamic and multifaceted. It involved not only medical adaptation but also identity negotiation, emotional regulation, and social positioning. This aligns with Havi Carel's phenomenological perspective, which frames illness as a transformation of bodily experience and a potential source of reflective awareness [16]

The findings support previous research suggesting that acceptance of chronic illness is associated with improved emotional and social well-being [17], and with a sense of harmony with oneself [18]. In this study, acceptance was often facilitated by sport, peer support, and self-management strategies, reflecting a broader psychosocial adaptation process [19].

Participants' ambivalence about being "ill" or "sick" shows the tension between medical labels and personal identity. This perspective resonates with recent work on narrative imagination and disability, which highlights the prevalence of 'supercrip' narratives and their implications for inclusion [20]. This echoes the work of Bright and Molnár [7], who showed how individuals with T1D use sport to renegotiate identity and resist stigmatization. The narratives collected here further demonstrate how ultra-endurance can serve as a space for reclaiming agency, reframing illness, and asserting a positive self-image.

## Specificities of ultra-endurance

Ultra-endurance sports represent a particularly demanding context for individuals living with type 1 diabetes. The physical and psychological challenges inherent to these practices intensify the mental load associated with diabetes management, requiring precise anticipation, continuous adaptation, and deep metabolic knowledge. These observations are consistent with broader research on ultra-endurance athletes, which highlights unique physiological demands and motivational profiles in extreme sports contexts [21]. Yet, paradoxically, this extreme context also offers a space of freedom, glycemic stability, and self-valorization.

Participants in this study described how ultra-endurance helped them develop advanced self-management skills, often acquired through trial and error and shared within peer communities. These findings align with Paley and Johnson's review [22], which identifies resilience strategies among ultra-endurance athletes—such as acceptance, emotional regulation, and positive reframing—that closely resemble those used by individuals managing chronic illness.

The narratives collected also illustrate how sport can serve as a medium for identity reconfiguration. Illness, rather than being solely a constraint, becomes a driver of self-transcendence and social visibility. This inversion of stigma is particularly evident in the way participants claim their condition, make it visible, and use it to inspire others. Pereira Vargas et al. [8] have shown that athletes can construct authentic identities by embracing vulnerability, provided they retain control over the narrative.

These findings add to the growing literature on how chronic illness intersects with performance and identity. They suggest that ultra-endurance is not merely a physical challenge, but a symbolic space where individuals negotiate autonomy, resilience, and recognition.

While the benefits of ultra-endurance practices are evident in terms of identity reconstruction and self-management, it is important to acknowledge that such extreme engagement may also carry risks. Previous research has highlighted potential issues such as physical overload, disordered eating, and exercise dependence among ultra-endurance athletes [23]. Although these aspects were not the focus of the present study, they warrant further investigation in future research.

## Strengths and limitations

This study provides rich insights into the lived experience of individuals with type 1 diabetes engaged in ultra-endurance sports. Its strength lies in the depth of the qualitative data, the reflexive positioning of the researcher, and the use of constructivist grounded theory to capture complex identity dynamics.

However, several limitations must be acknowledged. First, the sample size was small (n = 13) and composed of highly engaged athletes, which limits the generalizability of the findings. Although qualitative research does not seek statistical representativeness, the sample's homogeneity—especially in athletic commitment and peer-support involvement—may have amplified empowering narratives. Nevertheless, this size is consistent with qualitative standards and reflects the rarity of the target population: individuals living with type 1 diabetes and engaged in ultra-endurance sports are estimated to be fewer than one hundred in France. The gender imbalance (11 men, 2 women) mirrors the reality of ultra-endurance

participation, where women typically represent 15–25% of athletes. While this composition limits transferability to other contexts, it remains coherent with the population studied and allowed for rich, diverse narratives.

Finally, the dual role of the principal investigator as both a healthcare professional and a person living with type 1 diabetes may have influenced data collection and interpretation. While this insider position facilitated access and enriched understanding, it also raised potential risks of bias. Reflexive strategies—including maintaining a reflexive journal and engaging in regular debriefing with co-authors—were implemented to mitigate these risks and enhance credibility.

Second, recruitment through social media and peer networks may have introduced a selection bias, privileging participants already attuned to discourses of performance, autonomy, and resilience. This may have overshadowed more ambivalent or painful experiences, as documented in chronic illness research [24].

Third, the absence of comparison with other profiles of individuals living with T1D—such as non-athletes, those practicing other types of physical activity, or individuals in precarious social or medical situations—limits the scope of the conclusions. A comparative or longitudinal design could help refine the understanding of identity trajectories and adaptation strategies across diverse contexts.

Finally, the dual role of the principal investigator as both researcher and participant (living with T1D and practicing ultra-endurance) may have influenced data collection and interpretation. Although reflexive strategies were employed to mitigate this risk, including journaling and team triangulation, this insider position remains a methodological consideration.

## Practical implications

The findings of this study underscore the importance of recognizing the experiential knowledge developed by individuals living with type 1 diabetes, particularly those engaged in ultra-endurance sports. These individuals demonstrate advanced self-management skills, including anticipatory regulation, emotional coping, and personalized adaptation strategies. Such expertise, often acquired outside formal healthcare settings, represents a valuable resource for improving therapeutic adherence and patient empowerment [25,26].

For healthcare professionals, these insights call for a shift from a strictly biomedical model toward a more collaborative and patient-centered approach. This includes acknowledging patients as active partners in care, integrating their lived experience into clinical decision-making, and supporting autonomy through tailored education and dialogue [27].

Patient education programs (PEPs) should incorporate psychosocial dimensions, cultural sensitivity, and experiential learning. Several authors have emphasized the need for tailored therapeutic approaches that reflect patients' lived realities and self-management strategies [28]. The WHO recommends strengthening such programs to meet the challenges of chronic disease management, especially in contexts where patients must navigate their condition independently [29].

From a health policy perspective, the study highlights the structuring role of peer communities and physical activity in illness acceptance and identity reconstruction. These levers deserve institutional support through inclusive sports initiatives, urban planning that facilitates movement, and recognition of patient associations as legitimate actors in care governance [30–34].

Endocrinologists and general practitioners should actively discuss patients' ultra-endurance goals and offer tailored advice on insulin, nutrition, and risk management. Rather than discouraging such projects, clinicians can support safe participation by integrating experiential knowledge into therapeutic planning and fostering shared decision-making. General practitioners, who often maintain long-term relationships with patients, are ideally positioned to coordinate care, monitor overall health, and facilitate dialogue between specialists and patients pursuing extreme physical challenges.

## Future research directions

This study sheds light on an underexplored area: the subjective experience of chronic illness in the context of extreme sports. It invites further investigation into the intersections between ultra-endurance, identity, and self-management in type 1 diabetes.

Future research could adopt comparative designs, including individuals with T1D who engage in other forms of physical activity or those who do not practice sport, to better understand the diversity of adaptation trajectories. Studies involving non-athletic or socially vulnerable populations would help balance the empowering narratives observed here and reveal other forms of coping or constraint.

Longitudinal approaches would also be valuable to explore how illness identity evolves over time, in relation to life events, disease progression, and changing social contexts. Shneider et al.[3] have shown that illness identity is dynamic and closely linked to therapeutic adherence and health outcomes.

Finally, further research could examine the role of technological tools, peer communities, and healthcare relationships in shaping the experience of chronic illness. These dimensions, addressed in other components of the broader research project, deserve dedicated exploration to inform more inclusive and participatory models of care.

## Conclusion

This qualitative study, grounded in constructivist grounded theory and reported in accordance with COREQ guidelines, explored the lived experience of individuals with type 1 diabetes engaged in ultra-endurance sports. Through in-depth interviews and rigorous thematic analysis, the study revealed how illness, initially experienced as a biographical rupture, can become a space for adaptation, identity reconstruction, and resilience.

Ultra-endurance practices, while demanding, offer opportunities for self-reappropriation, social visibility, and advocacy. Participants developed advanced self-management strategies and reframed their condition as a source of strength rather than limitation.

The methodological approach—combining reflexive positioning, triangulation, and interpretive saturation—ensured analytical depth and credibility. While the findings are not statistically generalizable, they provide transferable insights into the psychosocial dynamics of chronic illness in performance-oriented contexts.

These results invite a rethinking of support models for chronic illness, emphasizing experiential knowledge, patient autonomy, and the role of peer communities. They also highlight the need for inclusive healthcare approaches that recognize diverse trajectories and forms of bodily engagement.

Ultimately, living with type 1 diabetes in ultra-endurance is not only about coping—it is about transforming the illness into a driver for going further, higher, stronger—differently.

## Author contributions

**Conceptualization:** Jean-Charles Vauthier, Bernard Kabuth.

**Formal analysis:** Jean-Charles Vauthier, Lucie Choley, Bernard Kabuth.

**Investigation:** Jean-Charles Vauthier.

**Methodology:** Jean-Charles Vauthier, Bernard Kabuth.

**Project administration:** Bernard Kabuth.

**Supervision:** Bernard Kabuth.

**Validation:** Lucie Choley.

**Writing – original draft:** Jean-Charles Vauthier.

**Writing – review & editing:** Lucie Choley, Bernard Kabuth.

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
