## [Decision Letter · Decision Letter 0]

18 Nov 2025

Dear Dr. VAUTHIER,

Thank you for submitting your manuscript to PLOS ONE. After careful consideration, we feel that it has merit but does not fully meet PLOS ONE’s publication criteria as it currently stands. Therefore, we invite you to submit a revised version of the manuscript that addresses the points raised during the review process.

We look forward to receiving your revised manuscript.

Kind regards,

Afagh Hassanzadeh Rad

Academic Editor

PLOS ONE

Journal Requirements:

“This research received no specific grant from any funding agency in the public, commercial, or not for-profit sectors.”

3. We note that your Data Availability Statement is currently as follows: “All relevant data are within the manuscript and its Supporting Information files. Interview transcripts are available from the corresponding author upon reasonable request, in accordance with ethical guidelines and participant consent. The study was approved by the South-East VI Research Ethics Committee (Reference: 24.00907.000253).”

4. In the online submission form you indicate that your data is not available for proprietary reasons and have provided a contact point for accessing this data. Please note that your current contact point is a co-author on this manuscript. According to our Data Policy, the contact point must not be an author on the manuscript and must be an institutional contact, ideally not an individual. Please revise your data statement to a non-author institutional point of contact, such as a data access or ethics committee, and send this to us via return email. Please also include contact information for the third party organization, and please include the full citation of where the data can be found.

6. We note you have included a table to which you do not refer in the text of your manuscript. Please ensure that you refer to Table 1 in your text; if accepted, production will need this reference to link the reader to the Table.

Reviewers' comments:

Reviewer's Responses to Questions

**Comments to the Author**

1. Is the manuscript technically sound, and do the data support the conclusions?

Reviewer #1: Yes

Reviewer #2: Yes

2. Has the statistical analysis been performed appropriately and rigorously?

Reviewer #1: Yes

Reviewer #2: Yes

3. Have the authors made all data underlying the findings in their manuscript fully available?

Reviewer #1: Yes

Reviewer #2: Yes

4. Is the manuscript presented in an intelligible fashion and written in standard English?

Reviewer #1: Yes

Reviewer #2: Yes

Reviewer #1: This qualitative study explores the lived experiences of individuals with Type 1 Diabetes (T1D) who participate in ultra-endurance sports, using a constructivist grounded theory approach. Based on 13 in-depth interviews, the authors identify four main themes: identity reconstruction, self-management, resilience, and social visibility. The paper provides meaningful insights into how living with a chronic condition can be transformed from a perceived limitation into a source of personal empowerment. The topic is original and highly relevant, addressing the intersection of chronic illness, identity, and extreme physical performance — an area that has received limited scholarly attention. The study is theoretically sound and methodologically coherent; however, several aspects could be improved to enhance clarity and rigor. Title: Please revise the manuscript title to remove the redundant word “Title.” Language and Style: A minor English language and style edit is recommended to improve overall fluency and readability. Sample Composition: The sample (n = 13) is relatively small and predominantly male (11 men, 2 women), which limits the transferability of findings. The authors should discuss the gender imbalance and its possible influence on the interpretations. Researcher Positionality and Bias: The primary author’s dual role as both a healthcare professional and a person living with Type 1 Diabetes warrants deeper reflexivity. Please elaborate on how this insider position may have shaped data collection and interpretation. There is a potential risk of bias related to the dual role of the main researcher, who is both a healthcare professional and a person living with Type 1 Diabetes. Comparative Context: The lack of comparison with other populations—such as non-athletes, individuals with varying physical activity levels, or those facing more complex sociomedical conditions—limits the interpretative scope of the study. Future comparative or longitudinal designs could provide deeper insight into the diversity of adaptation trajectories. Overall, this is an insightful and well-conducted qualitative study that contributes valuable perspectives to the literature on chronic illness and identity reconstruction. Addressing the above points would further strengthen the manuscript’s analytical depth and methodological transparency

Thank you again for the opportunity to review this work.

Sincerely

dr. matin mojaveri samak

Reviewer #2: Provide a dedicated table (e.g., Table 1: Participant Characteristics) with pseudonymized data: age, gender, years since T1D diagnosis, age at diagnosis, type/frequency of ultra-endurance events completed (e.g., UTMB, Diagonale des Fous, Ironman, 6-day races), glycemic control indicators if shared (e.g., recent HbA1c range, use of CGM/hybrid closed-loop/pump), and any notable complications or technology use.

Other relevant: Kellett & Winston (2021) on T1D runners' "supercrip" narratives; Thomas & Röhsli (2018) on adventure racing; or broader chronic illness/extreme sport studies (e.g., spinal cord injury ultrarunners).

Acknowledge potential positive bias .

Add clinical implications: How should endocrinologists respond when patients express ultra-endurance goals?

Consider slight refinement of title for precision: "Chronic Illness and Extreme Performance: How Ultra-Endurance Transforms

**Do you want your identity to be public for this peer review?** For information about this choice, including consent withdrawal, please see our Privacy Policy

Reviewer #1: **Yes: ** matin mojaveri samak

Reviewer #2: No

---

## [Author Response · Author response to Decision Letter 1]

21 Nov 2025

Dear Reviewers,

I would like to sincerely thank both reviewers for the time and effort devoted to reviewing my manuscript, which explores the lived experience of chronic illness—specifically type 1 diabetes—in the context of ultra-endurance sports. Your comments and suggestions have been extremely valuable and have significantly improved the quality of this work.

Revisions Implemented

• Title: The title has been revised to make it more precise and aligned with PLOS ONE style guidelines.

• Participant Table: The table has been enriched with additional data from interviews (age, sex, diabetes duration, technology use, type of race), while respecting ethical constraints. Race names were omitted to preserve anonymity.

• Ethics and Data Access: The ethics approval statement now appears only in the Methods section. The institutional contact for data access has been corrected (Data Access Committee, Université de Lorraine).

• References: The reference list was checked for retracted articles. Two suggested references could not be retrieved despite extensive searches. However, the concept of supercrip narratives was incorporated through the publication by Williams et al. (2021).

• Sample Justification: The sample size (n = 13) is consistent with qualitative research standards and reflects the rarity of the target population (likely fewer than one hundred ultra-endurance athletes with type 1 diabetes in France). The gender distribution (15% women) mirrors the reality of ultra-endurance events, where female participation remains limited but is gradually increasing.

• Reflexivity: I expanded the description of my dual role as both a physician and a person living with type 1 diabetes, and detailed strategies to minimize bias (reflexive journaling, triangulation with co-authors). This is addressed in the Methods section and acknowledged in the limitations.

• Clinical Implications: A paragraph was added to highlight how endocrinologists and primary care physicians can support patients pursuing ultra-endurance goals through individualized guidance and collaborative care.

• Language and Style: Several sentences were revised to improve clarity and readability, following your recommendation for a more fluent style.

To facilitate your review, a detailed point-by-point response table is provided ine the file "response to reviewers".

Once again, thank you for your constructive feedback. I hope this revised version meets your expectations and reflects the quality standards of PLOS ONE.

Best regards

---

## [Editor Report · Decision Letter 1]

25 Nov 2025

Chronic Illness and Extreme Performance: Type 1 Diabetes in Ultra-Endurance

PONE-D-25-52862R1

Dear Dr. VAUTHIER,

We’re pleased to inform you that your manuscript has been judged scientifically suitable for publication and will be formally accepted for publication once it meets all outstanding technical requirements.

Kind regards,

Afagh Hassanzadeh Rad

Academic Editor

PLOS ONE
---

## [Editor Report · Acceptance letter]

PONE-D-25-52862R1

PLOS One

Dear Dr. VAUTHIER,

I'm pleased to inform you that your manuscript has been deemed suitable for publication in PLOS One. Congratulations! Your manuscript is now being handed over to our production team.

Kind regards,

on behalf of

Dr. Afagh Hassanzadeh Rad

Academic Editor

PLOS One